# Neuroprotective Function of High Glycolytic Activity in Astrocytes: Common Roles in Stroke and Neurodegenerative Diseases

**DOI:** 10.3390/ijms22126568

**Published:** 2021-06-18

**Authors:** Shinichi Takahashi

**Affiliations:** 1Department of Neurology and Stroke, Saitama Medical University International Medical Center, 1397-1 Yamane, Hidaka-shi 350-1298, Japan; takashin@tka.att.ne.jp; Tel.: +81-42-984-4111 (ext. 7412) or +81-3-3353-1211 (ext. 62613); 2Department of Physiology, Keio University School of Medicine, 35 Shinanomachi, Shinjuku-ku, Tokyo 160-8582, Japan

**Keywords:** astrocyte, astroglia, glycolysis, human induced pluripotent stem cell, lactate, pentose-phosphate pathway

## Abstract

Astrocytes (also, astroglia) consume huge amounts of glucose and produce lactate regardless of sufficient oxygen availability, indicating a high capacity for aerobic glycolysis. Glycolysis in astrocytes is activated in accordance with neuronal excitation and leads to increases in the release of lactate from astrocytes. Although the fate of this lactate remains somewhat controversial, it is believed to fuel neurons as an energy substrate. Besides providing lactate, astrocytic glycolysis plays an important role in neuroprotection. Among the minor pathways of glucose metabolism, glucose flux to the pentose-phosphate pathway (PPP), a major shunt pathway of glycolysis, is attracting research interest. In fact, PPP activity in astrocytes is five to seven times higher than that in neurons. The astrocytic PPP plays a key role in protecting neurons against oxidative stress by providing neurons with a reduced form of glutathione, which is necessary to eliminate reactive oxygen species. Therefore, enhancing astrocytic glycolysis might promote neuronal protection during acute ischemic stroke. Contrariwise, the dysfunction of astrocytic glycolysis and the PPP have been implicated in the pathogenesis of various neurodegenerative diseases such as Parkinson’s disease, Alzheimer’s disease, and amyotrophic lateral sclerosis, since mitochondrial dysfunction and oxidative stress trigger and accelerate disease progression.

## 1. Introduction

Brain function is dependent on the oxidative metabolism of glucose. Therefore, a human adult brain weighing approximately 2% (1400 g) of the body weight (70 kg) is responsible for 25% of the total glucose consumption of the whole body [1,2]. Importantly, most glucose is oxidized by oxygen to generate carbon dioxide (CO_2_) and adenosine triphosphate (ATP). This high rate of oxidative metabolism of glucose cannot be replaced by any other energy substrates under normal physiological conditions [3,4,5]. Because glucose and oxygen are not stored in brain tissue, these essential substrates must be supplied by cerebral blood flow, and the cessation of a continuous supply causes irreversible cell damage within a short period of time [6,7,8]. Efficient ATP production is the basis of the generation of action potentials, and normal mitochondrial function in neurons is essential for this purpose. Unfortunately, however, high rates of oxidative metabolism of glucose always generate reactive oxygen species (ROSs) at a certain rate: 0.1–0.2% of the oxygen that is consumed is converted to ROSs [3,9]. In addition to mitochondria, peroxisomes are thought to be another important source of ROSs. Peroxisome is an organelle where very long chain fatty acids (>C22) are metabolized [10,11]. Although ROSs act as important signal molecules, in general, ROSs are regarded to play detrimental roles in cell injury [9,12,13,14]. Thus, several intrinsic protective mechanisms operate to reduce the toxic effects of ROSs in the brain. The dysfunction of these intrinsic mechanisms by which the brain eliminates ROSs leads to both acute [9,12,13,14] and chronic [15,16] neuronal damage. In ischemic stroke, for example, the elimination of ROSs during reperfusion therapy by either pharmacological thrombolysis or a mechanical thrombectomy is an essential therapeutic strategy, since damaged mitochondria during ischemia serve as a potential source of massive ROS production, especially after reperfusion. Thus, the intravenous administration of edaravone, a free radical scavenger, has been an established therapy for over 20 years [17,18]. Reactive oxygen species also play important roles in triggering both cytotoxic and vasogenic edema. Even with edaravone administration, formation of brain edema is sometimes unavoidable after recanalization therapy and massive brain edema results in death in the acute phase of cerebral infarction. Initial ischemia-induced cytotoxic edema, which is a reflection of energy failure, is followed by vasogenic edema that reflects an increased permeability of blood–brain barrier. In addition to ROS elimination, the development of effective treatment of brain edema is imperative [12,13,14,19,20,21,22]. Likewise, accumulated cellular damage caused by chronic ROS production might also be a potential mechanism of age-related neuronal damage, i.e., neurodegenerative diseases including Parkinson’s disease, Alzheimer’s disease, and amyotrophic lateral sclerosis (ALS). Recently, edaravone has also been added as a therapeutic option for the treatment of ALS [17,18]. Thus, enhancement of the intrinsic mechanism of ROS elimination seems to be a promising therapeutic strategy.

The high rate of glucose consumption in the brain is thought to mainly reflect neuronal glucose utilization. It is of note, however, that glucose consumption by glial cells, which outnumber neurons by a factor of 10, is not negligible, even though glial cells do not generate action potentials [23,24]. Among glial cells, astrocytes (or astroglia) are the most abundant glial cells in the brain, and their anatomical location allows astroglia to take up glucose directly from capillaries [23,24]. Astrocytes are interposed between neurons and capillaries. Astrocytic endfeet, therefore, play pivotal roles in the initiation of vasogenic edema through their water channel protein aquaporin-4 (AQP4) [21,22]. Neuronal synapses are enveloped by astrocytic endfeet to create tripartite synapses, and 99% of the capillary surface is also covered by the endfeet of astrocytes. In vitro data obtained using cultured rodent astroglia demonstrated that glucose utilization in astroglia is comparable with that in neurons, or even higher [3,4,5]. Importantly, glucose consumption by astrocytes may not consist of complete oxidation to generate CO_2_. In fact, astrocytes produce huge amounts of lactate even under a sufficient supply of oxygen (aerobic glycolysis). Their high glucose consumption and high glycolytic activity leads to lactate production, and lactate, in turn, is transferred to neurons to serve as a tricarboxylic acid (TCA) cycle substrate instead of glucose (astrocyte–neuron lactate shuttle model) [25,26]. Regardless of the long-lasting debate over this model, it is now widely accepted as an example of metabolic compartmentalization between astroglia and neurons [27,28,29,30,31,32,33]. One of the benefits of using lactate instead of glucose is that lactate enters directly into the TCA cycle after conversion to pyruvate, which generates 36 ATPs, while glucose needs to be metabolized to pyruvate through glycolysis and generates only 2 ATPs. Pyruvate is then transferred to mitochondria via the pyruvate dehydrogenase complex (PDHC) [1,2,3,4,5]. Both astrocytes and neurons possess monocarboxylate transporters (MCTs), and lactate is released from astrocytes and enters neurons through these transporters [3,4,5]. Besides lactate production, the astrocytic glycolytic pathway, per se, seems to have additional important roles [3,4,5]. More precisely, the pentose-phosphate pathway (PPP), a shunt pathway of glycolysis, plays a pivotal role in protecting neurons against oxidative stress [3,4,5,34,35,36]. This review will focus on the astrocytic PPP and its common beneficial roles in stroke and neurodegenerative diseases. Most of the data presented here were obtained using ^14^C-labeled tracer assays and cultured rodent astroglia and neurons (Figure 1) [3,4,5,37,38,39,40,41,42,43,44]. In addition to glucose metabolism, we also used ^14^C-labeled palmitic acid (long chain fatty acid; C16) and ^14^C-labeled β-hydroxybutyrate (BHB, a ketone body) to evaluate fatty acid oxidation and ketone body metabolism [42], respectively. We also discuss fatty acid metabolism in peroxisomes where very long chain fatty acids are metabolized.

## 2. Pentose-Phosphate Pathway in Astrocytes

The PPP, which is a minor pathway of glucose metabolism (contributing approximately 2–3%), is a shunt pathway in glycolytic metabolism [3,4,5,34,35,36]. After glucose is taken up by astrocytes via a glucose transporter, the first reaction in glycolysis, the conversion from glucose to glucose 6-phosphate (G6P), is catalyzed by hexokinase. The PPP branches at G6P with an initiation by glucose 6-phosphate dehydrogenase (G6PDH), a rate-limiting enzyme of PPP flux. In the PPP, a main product is nicotinamide adenine dinucleotide phosphate (NADPH), a cytosolic reducing equivalent, and NADPH is utilized to convert the oxidized form of glutathione (GSSG) back to the reduced form of glutathione (GSH). In fact, GSH is a major antioxidative that eliminates ROSs in cooperation with glutathione peroxidase [45,46,47,48]. Thus, PPP flux is thought to be an indicator of the cellular antioxidative capacity.

The quantitative measurement of glucose flux into the PPP was evaluated using 1-^14^C- and 6-^14^C-labeled glucose in both cultured neurons and astroglia prepared from rats or mice (Figure 1) [3,4,5,41,43,44]. Despite the comparable glucose uptake and phosphorylation (measured using ^14^C-labeled deoxyglucose) by hexokinase in astroglia, compared with neurons, the flux into the astroglial PPP was five to seven times higher than that into the neuronal PPP [3,4,5,41,43,44]. A high glucose flux into the PPP is, at least in part, a reflection of the high glycolytic rate in astroglia. In the resting state, astrocytic glucose utilization is comparable with that of neurons. Even though neurons are not in direct contact with capillaries, glucose supplied from capillaries reaches neurons through the extracellular space. In fact, neurons possess glucose transporters and uptake glucose avidly. Importantly, neuronal excitation enhances glucose utilization in astrocytes, rather than in neurons, since glutamate released from excited neurons is taken up by astrocytes and this, in turn, activates astrocytic glucose utilization (Figure 2) [3,4,5,25,26]. These issues are further discussed later in this review. 

Astrocytes contain functional mitochondria in terms of the oxidative metabolism of glucose. Regardless of the potentially active oxidative metabolic capacity in astroglial mitochondria, the catabolism of lactate/pyruvate in astroglia is much lower than in neurons. For example, ^14^CO_2_ production from ^14^C-labeled glucose by cultured astroglia, corrected according to the cellular protein content, was approximately 1/6–1/4 of that produced by cultured neurons [3,4,5,37,38]. Similarly, ^14^C-labeled pyruvate or ^14^C-labeled lactate produced by cultured astroglia was also 1/6–1/4 of that produced by cultured neurons [3,4,5,37,38]. In contrast, however, the oxidation of ^14^C-labeled glutamate and the resultant production of ^14^CO_2_ were much higher in astroglia [5], suggesting a potential mitochondrial oxidative capacity (Figure 1). These observations also suggest that glutamate, rather than glucose, can serve as an energy substrate of the astrocytic TCA cycle when glutamate is taken up by astrocytes after neuronal release. Again, the oxidation of ^14^C-labeled glucose or lactate and the resultant production of ^14^CO_2_ are negligible in astroglia [3,4,5,37,38]. As for neurons, both ^14^C-labeled glucose and lactate are well oxidized to produce ^14^CO_2_ [3,4,5,37,38]. It seems reasonable to speculate that lactate produced in astroglia is transferred to neurons, where it is used as a substrate for the mitochondrial TCA cycle to produce ATP (astrocyte–neuron lactate shuttle model). More interestingly, competition assays have demonstrated that lactate does indeed seem to be the favored substrate for neurons, compared with glucose [3,4,5,37,38]. 

Astrocytic glucose utilization seems to occur in order to fuel the PPP flux and produce lactate [3,4,5]. In contrast, neuronal glucose utilization seems to have a rather limited role, especially under functional activation. Functional activation reportedly induces transient lactate overproduction in vivo [3,4,5]. In fact, the cerebral metabolic rate of oxygen (CMRoxy)/cerebral metabolic rate of glucose (CMRglc), which is normally 6, is indicative of the complete oxidation of glucose in the resting state; this rate decreases to 5 during functional activation [3,4,5]. Neurons seem to utilize lactate instead of glucose for efficient ATP production at the sacrifice of glycolysis and its antioxidative roles [3,4,5]. Since the antioxidative capacity of neurons is not sufficient given their high rate of oxidative metabolism, astrocytes support neurons by providing GSH, taking advantage of the high PPP flux that is associated with their high glycolytic activity (Figure 2) [3,4,5].

Glutathione synthesis is also reported to be more active in astrocytes than in neurons [3,4,5]. Glutathione in astrocytes is released into the extracellular space to reduce ROSs, and the released GSH is then hydrolyzed by the γ-glutamyltranspeptidase present on the external surfaces of astrocytes, producing the dipeptide cysteinyl-glycine that is then hydrolyzed by aminopeptidase N to release cysteine and glycine [3,4,5]. These amino acids, along with glutamine released from astrocytes, provide all the necessary precursors for neuronal GSH synthesis (Figure 2).

## 3. Activation of Astrocytic Pentose-Phosphate Pathway under Altered Supply of Glucose

Our quantitative measurement of basal PPP demonstrated that PPP flux in astroglia is approximately five to seven times higher than in neurons [3,4,5]. The effects of a high glucose environment on changes in PPP flux is another important issue. Diabetes mellitus serves as a major risk for cognitive impairment as well as cerebral vascular diseases affecting both small and large arteries [3,4,5,37,41,49]. Since oxidative stress associated with a hyperglycemic state has been implicated in the underlying mechanism of the injury of brain vessels as well as brain cells, the enhancement of defense mechanisms against oxidative stress seems to be important for preventing injury. We evaluated the effect of altering the glucose concentration on glucose metabolism in both neurons and astroglia [37,41,49]. A normal glucose concentration in plasma is 5 mmol/L (90 mg/dL) in the fasting state. Importantly, however, the concentration of glucose in the extracellular space in the brain is 2–3 mmol/L and increases as the plasma glucose concentration increases [1,2,3,4,5,37,41,49]. As stated above, glycolytic activity in cultured astroglia is higher than in cultured neurons grown at a near normal glucose concentration in the brain (5 mmol/L). We found that increasing concentrations of glucose in cultured medium for both short [41] and longer [37] periods of time enhanced glycolytic activity in astroglia. First, the chronic exposure of astroglia to a high glucose concentration increased the glycogen content in astroglia and their glycolytic activity [37], suggesting the enhancement of PPP flux and the antioxidative capacity of astrocytes in diabetic patients [41]. Second, a fluctuating glucose concentration in the culture medium also enhanced the glycolytic activity in astroglia [49]. Finally, an acutely increasing concentration of glucose during a PPP flux assay showed that the PPP flux in astroglia increased more steeply with rising concentrations of glucose than it was observed to in neurons. These results suggest that the astroglial PPP plays an antioxidative role in response to both acute and chronic high glucose environments as well as fluctuating glucose concentrations [37,41,49].

## 4. Effects of Brain Ischemia on Astroglial PPP

As expected, a hypoxic condition enhances glycolytic metabolism because of the lack of oxygen [39]. Since oxygen extraction from arterial blood is relatively higher (30%) than glucose extraction (only 10%), ischemia caused by the occlusion of cerebral vessels mimics a hypoxic state where the glucose supply is relatively maintained. Of note, glycogen storage, observed exclusively in astroglia, enables astrocytes to undergo glycolysis even without an oxygen supply. Using a hypoxic chamber, cultured astroglia and neurons were exposed to 1% O_2_ for 12 or 24 h. As expected, high levels of lactate production were observed in both neurons and astroglia [39]. Importantly, however, the changes in the PPP flux in both types of cells contrasted sharply. The PPP flux in cultured astroglia showed a marked elevation after hypoxia, while it decreased in cultured neurons [39].

The enhanced flux in the PPP observed in hypoxic astroglia could be of value after reperfusion. Spontaneous recanalization after large vessel occlusion in ischemic stroke occurs at a rate of 10–20%, and recanalization therapy using alteplase or combined with a mechanical thrombectomy dramatically improves the recanalization rate (80–90%) [50]. As a result, neurons in danger of irreversible damage and cell death can be rescued. A major problem that should be solved is the efficient elimination of ROSs that are produced in association with reperfusion [51,52,53,54]. Reactive oxygen species damage not only neuronal cells in danger, but also vascular endothelial cells. They activate the matrix metalloproteinase (MMP) system, resulting in the degradation of the extracellular matrix (ECM). The detachment of astrocytic endfeet from the ECM and loosened tight junctions between the endothelium lead to vascular damage and hemorrhagic transformation [51,52,53,54]. Astrocytes are interposed between neurons and capillaries, and this position makes them suitable for the protection of both neurons and brain vessels under the reperfusion state after stroke intervention [39,54].

Astrocytic water channel protein, AQP4, plays a crucial role in the generation of both cytotoxic and vasogenic edema [21,22]. Hypoxia/ischemia per se, as well as ROSs produced after recanalization, induce AQP4 expression on astrocytic endfeet which leads to brain edema. A massive brain edema is sometimes lethal and the management of brain edema is an important issue in cerebral infarction, as well as in traumatic brain injury. Pharmacological inhibition, as well as genetic manipulation of AQP4, can reduce brain edema substantially [21,22]. Interestingly, it remains to be determined whether or not hypothemia is beneficial for brain edema [55,56,57]. In fact, the effects of hypothemia on astrocytic protective roles through glucose metabolism seem to be complex, because hypothemia suppresses their glucose metabolism, which is thought to be neuroprotective, as described above [55,56,57]. 

## 5. Glutamate Released from Neurons Triggers Increased PPP Flux in Astroglia 

Functional activation in the brain is mediated by various neurotransmitters. In particular, glutamate is a ubiquitous excitatory neurotransmitter that is distributed throughout the brain. The excessive release of glutamate triggers neuronal injury, i.e., excitatory cell death, and this mechanism has been implicated as a main component in the pathogenesis underlying ischemic neuronal cell death, as well as neurodegenerative diseases such as ALS [58,59]. In addition to glutamate toxicity, the hyperexcitation of motor neurons per se induces oxidative stress leading to mitochondrial dysfunction and eventual cell death [58,59]. To prevent glutamate-induced excitotoxicity, glutamate must be taken up by astroglial endfeet that envelope the synapse immediately after its release into the synaptic cleft. This uptake of glutamate is achieved by Na^+^-dependent glutamate transporters using the Na^+^ gradient across the astroglial cellular membrane. To restore the Na^+^ gradient, Na^+^, K^+^-ATPase consumes ATP that is generated in astroglia mainly through glycolysis using the glucose or TCA cycle using glutamate as substrates [4,5,25,26]. When PPP flux in astroglia was measured before and after glutamate exposure, robust increases in the astroglial PPP flux, as well as glucose utilization, were observed, indicating that neuronal excitation activates the PPP flux that serves as an antioxidative defense mechanism in astroglia (Figure 2) [4,5,44].

As described above, ATP production in astrocytes induced by glutamate can also be fueled by glutamate itself. Glutamate is converted to α-ketoglutarate and serves as a TCA cycle substrate. Although most glutamate taken up by astrocytes is recycled back to neurons (glutamate–glutamine shuttle) [4,5], some glutamate might serve as an energy substrate in astroglia [3,4,5]. Glutamate uptake is dependent on Na^+^, K^+^-ATPase, and ATP production in astrocytes seems to rely on glycolysis [3,4,5,25,26]. Astroglia exposed to glutamate exhibit enhanced glucose utilization in a Na^+^-dependent manner, followed by increased lactate production [5] and an enhanced PPP flux [44]. However, whether ATP production through glycolysis alone is sufficient to maintain glutamate uptake remains unclear. Glutamate-derived α-ketoglutarate could also be utilized to produce ATP through the astrocytic TCA cycle in mitochondria (Figure 2). We measured ^14^CO_2_ originating from ^14^C-labeled glutamate and found that glutamate oxidation increases in a dose-dependent manner, suggesting that glutamate taken up by astroglia can serve as an energy substrate [5]. Interestingly, glutamate can also be a source of lactate production by astroglia [5]. Astrocytic malic enzyme generates glutamate-derived lactate, which may also be exported into the extracellular space and serve as a TCA cycle substrate in neurons [5].

## 6. Fatty Acid Oxidation under Hypoxia and Reoxygenation 

Fatty acid metabolism in astroglia seems to be an important issue [4,5,42]. As mentioned in the introduction, peroxisomes where very long chain fatty acids are β-oxidized are thought to be a source of ROSs [10,11]. In astrocytic peroxisomes, not only very long chain fatty acids but also long chain fatty acids are metabolized. The astrocytic peroxisomes are important for the maintenance of myelin structure in brain. We characterized the metabolism of long chain fatty acid using ^14^C-labeled palmitic acid in cultured neurons and astroglia [4,5,42]. Although both types of cells metabolized ^14^C-palmitic acid through β-oxidation, astroglia produced a large amount of BHB (i.e., a major ketone body). Moreover, ketone body production by astroglia was activated under a hypoxic/hypoglycemic state through the action of adenosine monophosphate-dependent kinase (AMPK). 

Ketone bodies can be a source of cholesterol synthesis as well as energy substrates for the TCA cycle [10,11]. Therefore, we hypothesized that ketone bodies produced by astrocytes are transferred to neurons serving as an energy substrate for their TCA cycle. Ketone bodies are metabolized to acetyl-CoA and enter the TCA cycle. Therefore, both lactate and ketone bodies can serve as TCA cycle substrates [3,4,42]. Importantly, however, conversion of ketone bodies to acetyl-CoA does not need PDHC. This is a beneficial aspect of BHB, as compared with lactate, under ischemia/reperfusion. Since PDHC is susceptible to ROSs, neuronal oxidative metabolism by mitochondria cannot be restored even after reperfusion, despite accumulated lactate during the ischemic state. In contrast to lactate, BHB can be a TCA cycle substrate even when PDHC is dysfunctional. In regards to ROS production in association with ketone body production through β-oxidation in peroxisomes or mitochondria, a strong anti-oxidative system in astrocytes may allow them to protect themselves [4,5,42].

An agonist of peroxisome-proliferator activated receptor γ (PPARγ) is thought to be neuroprotective [60]. We evaluated the effect of pioglitazone, a PPARγ agonist, on glucose metabolism using cultured astroglia and neurons. Pioglitazone activated astroglial glycolysis and lactate production, while it enhanced neuronal oxidative metabolism of lactate in a protein kinase-A (PKA)-dependent manner [61]. The exact location where pioglitazone exerted the metabolic effect remains to be elucidated. It has been reported that PPARγ agonists enhance astrocytic catalase activity, which erases ROSs produced in peroxisomes [11]. Thus, peroxisomes, mitochondria, and cytosolic interaction facilitate neuronal energy metabolism via production of astrocytic energy substrates. 

## 7. Activation of Pentose-Phosphate Pathway with Dopamine Metabolism

Dopamine is a unique neurotransmitter that is employed in the nigro-striatal system. The degeneration of dopaminergic neurons in the substantia nigra and the resultant deficiency in dopamine release in striata causes Parkinson’s disease. The precise mechanism by which dopaminergic neurons degenerate, especially in idiopathic Parkinson’s disease without any apparent mutation in mitochondrial metabolism, remains to be solved [44,59,62]. One of the hypothetical mechanisms is oxidative stress induced by the continuous activity of dopaminergic neurons. In fact, dopaminergic neurons in the substantia nigra exhibit a pace-making activity at 2–4 Hz to maintain the dopamine concentration in the striata at a certain level (Figure 3) [44,59,62,63,64]. This activity, in turn, causes oxidative stress in the mitochondria of dopaminergic neurons. In fact, the mitochondria in dopaminergic neurons require constant repair or replacement by their maintenance system, and genetic abnormalities in this system are the major pathogenetic mechanisms of several familial Parkinson’s diseases. 

Unlike other neurotransmitters, dopamine has its own characteristics that induce oxidative stress. Dopamine released from the nerve terminals of dopaminergic neurons into the synaptic cleft is autoxidized to generate dopamine quinone, which also acts as a source of oxidative stress (Figure 3) [44,59,62,63,64]. Unlike glutamate, dopamine recycling is dependent on the dopaminergic neurons themselves via Na^+^-dependent dopamine transporters. Therefore, once the degeneration of dopaminergic neurons has been initiated, dopaminergic recycling is also impaired and accelerates the subsequent damage. Astrocytes are capable of taking dopamine up through dopamine transporters as well as serotonin transporters, both of which are Na^+^-dependent. When some dopamine is taken up by astroglia using the Na^+^-gradient maintained by astroglial Na^+^, K^+^-ATPase, ATP production is triggered and astrocytic glycolysis is activated. Therefore, the exposure of astroglia to dopamine may also increase the PPP flux in astroglia [44].

## 8. Regulation of PPP Flux by G6PDH Transcription by Nrf2

Reactive oxygen species, as well as dopamine, may act as PPP activators at the transcriptional level. G6PDH is a rate-limiting enzyme of PPP flux at G6P [3,4,5,39,40,41,43,44]. This enzyme is regulated by both an allosteric mechanism and a transcriptional mechanism. In the latter mechanism, the Kelch-like enoyl-CoA hydratase-associated protein 1 (Keap1)/nuclear factor erythroid 2 p45 subunit-related factor 2 (Nrf2) system plays a pivotal role (Figure 4) [3,4,5,39,40,41,43,44]. Nrf2 is a transcriptional factor that translocates into the nucleus when triggered by danger signals and various detoxifying enzymes, such as glutathione peroxidase and glutathione S transferase, as well as G6PDH. Under normal resting conditions, Nrf2 is anchored with its adaptor protein Keap1 and forms a heterodimer. The Keap1/Nrf2 complex is constantly being degraded by the ubiquitin–proteasome system; thus, the transcriptional activity of Nrf2 is inhibited in normal, unstressed brain. 

Reactive oxygen species attack the thiol of the cysteine residue of Keap1 and induce conformational changes, releasing Nrf2 from Keap1. Free Nrf2 molecules translocate into the nucleus and initiate the transcription of target genes by binding to the antioxidant response element (ARE) (Figure 4). After hypoxic conditions, reoxygenation induces robust ROS production that, in turn, activates the defense mechanism of the astrocytic PPP via a transcriptional mechanism. An enhancer of the Keap1/Nrf2 system, therefore, could act as a neuroprotective drug that functions via astrocytic PPP activation [65,66,67,68,69,70]. A classical Nrf2 activator, sulforaphane, demonstrated a marked enhancement of PPP flux and resultant increases in GSH content in cultured astroglia [41]. Dimethyl fumarate (DMF), which is now in clinical use as a disease-modifying drug for multiple sclerosis, is thought to act as a Nrf2 activator [65,66,67]. Another Nrf2 activator, bardoxolone methyl [68,69,70], has also exhibited therapeutic potential for the treatment of kidney diseases. Unfortunately, however, this drug does not cross the blood–brain barrier and therefore cannot activate the PPP in astrocytes in the brain. 

Another mechanism by which the Keap1/Nrf2 system is activated is through the phosphorylation of Nrf2. Several kinases are reported to phosphorylate serine residues of the Nrf2 protein, leading to Nrf2-initiated transcription (Figure 2) [41]. High glucose environments can activate the hexosamine biosynthetic pathway (HBP), which is another minor pathway of glucose metabolism. An end-product of HBP, N-acetylglucosamine, induces the activation of endoplasmic reticulum (ER) stress transducer protein kinase RNA (PKR)-like ER kinase (PERK), which phosphorylates Nrf2 (Figure 2) [41]. Phosphorylated Nrf2 is released from Keap1 and translocates into the nucleus, leading to the transcription of astrocytic antioxidative enzymes. The possibility of ER stress-induced PPP activation should be examined in diabetic patients [41].

## 9. Dopamine-Induced Activation of Astroglial PPP

The mechanism of the dopamine-induced activation of PPP in astroglia seems to be more complex. As described above, dopamine itself acts as an origin of ROS [44]. In addition, the pace-making activity of dopaminergic neurons is also a potential origin of mitochondrial overload, resulting in ROS production [71]. According to our observations, the extent of glycolytic activation in astroglia induced by dopamine uptake is much weaker than that induced by glutamate [44]. Therefore, protecting dopaminergic neurons against oxidative stress is insufficient. We examined the role of the astroglial Keap1/Nrf2 system in protecting neurons against dopamine-induced oxidative stress. Rodent primary cultures of striatal neurons exhibited cell damage after exposure to dopamine for 36–48 h, while cocultures with astroglia protected the neurons from cell damage [44]. Interestingly, cultured astroglia prepared from Nrf2 knockout mice did not confer any protective effects when cultured with neurons from wild-type mice, suggesting that astroglial Nrf2-dependent neuroprotection plays an important role. In fact, exposure to dopamine for 15 h both activated the PPP flux and increased the GSH content in astroglia [44]. 

In addition to protective roles for dopaminergic neurons, astrocytes per se can potentially contribute to restoring dopaminergic neurons lost in Parkinson’s disease. Recent advancements in genetic engineering have enabled transdifferentiation of astrocytes in the midbrain into dopaminergic neurons. Astrocyte-derived dopaminergic neurons can restore nigrostriatal pathways and work for longer periods of time under the protection of astrocytes [72]. 

## 10. Toll-Like Receptor 4 and Keap1/Nrf2 under Ischemia

The roles of the inflammatory response in cellular injury after ischemia are a focus of research. The immediate response to energy failure within 24 h after the onset of cerebral ischemia includes oxidative stress and glutamate toxicity, as described above. These early responses are followed by neuroinflammation mediated by glial cells; toll-like receptor 4 (TLR4) plays a central role in this process [43,73,74]. Toll-like receptor 4, which mediates the innate immune system, is expressed in microglia and astroglia in the central nervous system (Figure 5). In addition to lipopolysaccharide (LPS), which is a well-known ligand of TLR4, damaged mitochondria are also a natural ligand of LPS, suggesting that mitochondrial fragments generated during the early events of cerebral ischemia in turn act as ligands for TLR4 and induce secondary inflammation. In addition to mitochondria, several molecules originating from damaged brain structure compose “damage-associated molecular patterns” (DAMPs). These endogenous ligands activate TLR4 in microglia and components of the down-stream injury pathway such as myeloid differentiation protein 88-nuclear factor-κB (Myd88-NFκB), mitogen-activated protein kinase (MAPK), and Janus kinases/signal transducers and activators of transcription (JAK/STAT). The activation of these pathways leads to inducible nitric oxide synthase (iNOS) and NADPH oxidase (NOX), which generate reactive nitrogen species (RNSs) and ROSs, respectively. Both RNSs and ROSs induce further cell damage during cerebral ischemia.

As described above, ROSs are potentially injurious to cells, but they also exert protective effects against themselves by inducing the antioxidative system in astrocytes [43]. We evaluated the effect of TLR4 activation on astrocytic PPP flux in cultured astroglia after LPS exposure. First, LPS exposure induced ROS production, but not nitric oxide (NO) production, in cultured astroglia. The application of hydrogen peroxide (a ROS) to astroglia induced huge increases in the PPP flux, suggesting the ROS-mediated activation of the Keap1/Nrf2 system and the resultant enhancement of PPP flux. As noted before, hypoxia per se induces astroglial PPP activation [39]. When cultured astroglia were incubated in a hypoxic chamber with LPS, PPP increases were observed in an additive manner [40]. PPP activation is thought to be mediated by the activation of glycolysis during the immediate-early phase and by the induction of the rate-limiting enzyme G6PDH during the subsequent phase in a coordinated manner [43]. 

Moreover, extracellular-signal regulated kinase 1/2 (ERK1/2), a type of MAPK, plays an important role in the regulation of PPP flux. During the activation of the TLR4-triggered neuroinflammatory response, the activation of MAPK phosphorylates various substrates, including Nrf2. Namely, the serine or threonine residues of Nrf2 are phosphorylated by an MAPK (in this case, ERK1/2), resulting in the release of Nrf2 from its Keap1 anchor and allowing Nrf2 to translocate into the nucleus and induce the transcription of G6PDH and GSH-related enzymes (Figure 4). The LPS-induced activation of PPP flux in astrocytes may partly depend on this pathway.

## 11. Interaction between Microglia and Astroglia

So far, this review has focused on neuronal–astroglial interactions (Figure 2 and Figure 3) [3,4,5]. Interactions between microglia and astroglia are another attractive subject, especially with regard to the debate over the detrimental or beneficial roles of glial cells [75,76,77,78,79,80]. Astroglia are thought to act as an enhancer of neuroinflammation [80,81]. In particular, several proinflammatory cytokines released from microglia trigger not only direct neuronal damage, but also induce astrocytic release of neurotoxic cytokines. In these sequential events, TLR4 and its ligand are key players in this process. Ischemic insults cause neuronal damage and generate DAMPs, which act as ligands for TLR4. Microglia express TLR4 abundantly, while astroglial TLR4 expression seems to be limited. Lipopolysaccharide, a natural TLR4 ligand, induces ROS and NO, which are thought to be neurotoxic, production in microglia (Figure 5) [43]. In addition, microglia release pro-inflammatory cytokines upon TLR4 stimulation and these cytokines induce a neurotoxic phenotype in astroglia. On the other hand, some microglial signals induce in astroglia a neuroprotective phenotype. Recently, Shinozaki et al., reported that microglia lower P2Y receptor expression in astrocytes, inducing a neuroprotective phenotype [76]. We also found that NO released from microglia activated astroglial PPP flux through the Keap1/Nrf2 system. Namely, the NO-induced nitrosylation of Keap1 residues released Nrf2 from Keap1, allowing Nrf2 to act as a transcription factor in an in vitro model of cultured rodent microglia and astroglia (Figure 5) [43]. Thus, the balance between two signals that direct astrocytes toward opposite functions seems to be important.

The limitations of cultured cells should be kept in mind. An essential question is whether rodent cells are a suitable model for human neural cells. To address this issue, induced pluripotent stem cell (iPSC) technology appear to be a powerful tool. We recently showed that the astrocyte–neuron lactate shuttle is also feasible in human astrocytes and neurons using human iPSC derived astroglia and neurons [82]. Likewise, human iPSC-derived microglia and astroglia would enable more reliable information regarding their possible interactions to be obtained. Furthermore, to investigate an interaction between astrocytes and microvessels, flow dynamics should be taken into consideration [83,84]. Eventually, a 3-dimensional organ model will elucidate the detailed mechanism by which astrocytes regulate microcirculation [85] and cerebral organoids can also be used as therapeutic tools in the future [86].

## 12. Conclusions and Future Perspectives

This review discussed the roles of astrocytic glycolysis in neuroprotection. The high glycolytic activity of astrocytes, especially during neuronal activation, leads to lactate production and PPP activation. The former serves as an appropriate energy substrate for energy-consuming neurons. The latter produces NADPH, a reduced equivalent, and leads to a powerful antioxidative material, GSH. These features allow neurons to dedicate their energy production to the generation of action potentials. The dysfunction of astrocytic glycolysis and the PPP, therefore, leads to neuronal dysfunction in acutely progressive diseases, such as stroke, as well as in chronically progressive neurodegenerative diseases. The activation of astrocytic endogenous protective mechanisms could provide future therapeutic strategies for various neurological disorders.

This review did not discuss the third type of glial cell, oligodendrocytes. The reported model for interactions among neurons, capillaries, and astrocytes is applicable only in the gray matter of the brain. In white matter, the neuronal axons are myelinated by oligodendrocytes [87,88,89,90]. Thus, a capillary–astrocyte–oligodendrocyte–neuron model should be considered. Regarding the lactate supply to axons, oligodendrocytes, rather than astrocytes, are thought to act as suppliers in white matter [85,90,91,92,93]. If this is true, the glycolytic activity of oligodendrocytes should be evaluated in greater detail. Nevertheless, capillaries are covered by astrocytic endfeet, and glucose flux from astrocytes to oligodendrocytes as well as neurons should be a future target of study. To develop new drugs that facilitate astrocytic protective function, iPSC-based drug repositioning, as well as computer-assisted drug screening, will be a powerful tool [91,92,93].

## Figures and Tables

**Figure 1 ijms-22-06568-f001:**
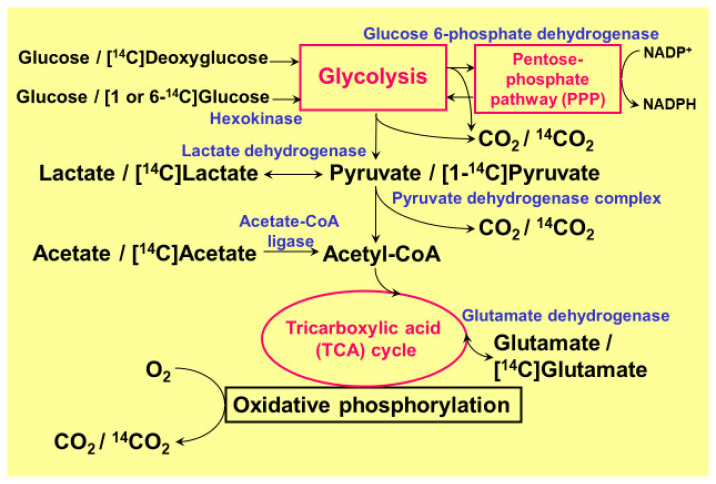
Pathways of glucose metabolism and ^14^C-tracers used in quantitative metabolism. The main pathway of ATP generation from glucose in the brain, especially in neurons, depends on oxidative phosphorylation in the tricarboxylic acid (TCA) cycle. In astrocytes, however, glycolysis dominates, and glucose flux into its shunt pathway, the pentose-phosphate pathway (PPP), is five to seven times higher than in neurons. The PPP plays an important role in generating a reduced form of nicotinamide adenine dinucleotide phosphate (NADPH) from the oxidized form (NADP^+^). Although the oxidation of [^14^C] lactate or [^14^C] pyruvate and the production of ^14^CO_2_ is much lower (1/6–1/4) in astroglia than in neurons, astroglial [^14^C] glutamate oxidation is much higher, suggesting normal mitochondrial function.

**Figure 2 ijms-22-06568-f002:**
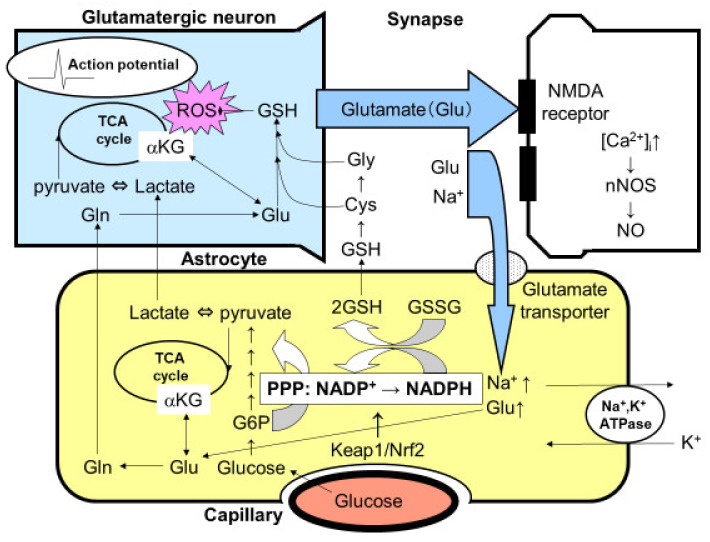
Tripartite synapse consisting of glutamatergic neurons and astrocytic endfoot. Glutamate (Glu) is a ubiquitous excitatory neurotransmitter that causes cell death through the N-methyl-D-aspartate (NMDA) receptor-mediated intracellular Ca^2+^ concentration ([Ca^2+^]_i_). The elevation of [Ca^2+^]_i_ activates neuronal nitric oxide synthase (nNOS) leading to the production of nitric oxide (NO), which is a major cell death mediator of neurons. Released glutamate, therefore, must be taken up by astrocytic Na^+^-dependent glutamate transporters. Glutamate uptake stimulates astrocytic Na^+^, K^+^-ATPase, thereby enhancing glycolytic metabolism (aerobic glycolysis) in astrocytes. The increased lactate/pyruvate, end-products of glycolysis, then fuel the neuronal tricarboxylic acid (TCA) cycle as energy substrates (astrocyte–neuron lactate shuttle model). Mitochondria are a major source of reactive oxygen species (ROS), which injure neurons both acutely and chronically, in the central nervous system. Glutamate in astrocytes is converted into glutamine (Gln) and then recycled back to glutamatergic neurons (glutamate–glutamine cycle). Glu can also serve as an energy substrate, α-ketoglutarate (αKG). The activation of glycolysis in astrocytes increases flux into the pentose-phosphate pathway (PPP), a shunt pathway of glycolysis, at glucose 6-phosphate (G6P). In addition to allosteric regulation, the transcriptional regulation of the rate-limiting enzyme of PPP, G6P dehydrogenase, is under the Kelch-like enoyl-CoA hydratase-associated protein 1 (Keap1)/nuclear factor erythroid 2 p45 subunit-related factor 2 (Nrf2) system. Increased flux to the PPP increases the ratio of the reduced form and the oxidized form of nicotinamide adenine dinucleotide phosphate (NADPH/NADP^+^), which is used to convert the oxidized form of glutathione (GSSG) to the reduced form of glutathione (GSH). Glutathione can be transferred to neurons via Gln, cysteine (Cys), and glycine (Gly).

**Figure 3 ijms-22-06568-f003:**
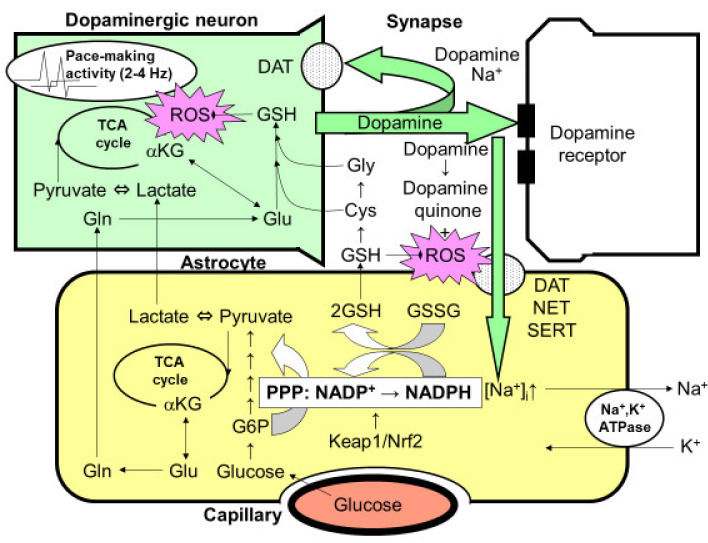
Possible interaction between astrocytes and dopaminergic nerve terminals. The optimal dopamine concentration is maintained in the striatum by the 2–4-Hz pace-making activity of dopaminergic neurons in the substantia nigra. However, this constant activity can elicit mitochondrial overload during oxidative metabolism, leading to the increased production of reactive oxygen species (ROSs). The reduced form of glutathione (GSH) plays an important role in eliminating ROSs. The synthesis of glutathione requires glutamine (Gln)/glutamate (Glu), which is supplied by astrocytes, as well as cysteine (Cys) and glycine (Gly), which are also derived from GSH released from astrocytes. Thus, neuronal GSH synthesis is largely dependent on astrocytes. After dopamine is released from the presynaptic terminals of dopaminergic neurons, it is taken up by Na^+^-dependent dopamine transporter (DAT) expressed in presynaptic neurons. However, the astrocytes that surround the synapses also seem to take up dopamine via two astrocytic Na^+^-dependent monoamine transporters: norepinephrine transporter (NET) and serotonin transporter (SERT). Increases in the intracellular Na^+^ concentration ([Na^+^]_i_) activate Na^+^, K^+^-ATPase, thereby enhancing glycolytic metabolism (aerobic glycolysis) in astrocytes. The increased lactate/pyruvate, end-products of glycolysis, then fuel the neuronal tricarboxylic acid (TCA) cycle as energy substrates. In astrocytes, the pentose-phosphate pathway (PPP), a shunt pathway of glycolysis, is co-activated; this process facilitates the conversion of NADP^+^ to NADPH. NADPH is necessary for glutathione reductase to eliminate ROSs, since glutathione peroxidase requires the reduced form of GSH converted from the oxidized form of glutathione (GSSG), which is dependent on NADPH. GSH released from astrocytes also plays an important role in diminishing dopamine-derived ROSs, since dopamine produces dopamine quinone and ROSs via auto-oxidation in the extracellular space. A rate-limiting enzyme of the PPP, glucose 6-phosphate (G6P) dehydrogenase, is regulated transcriptionally by the Kelch-like enoyl-CoA hydratase-associated protein 1 (Keap1)/nuclear factor erythroid 2 p45 subunit-related factor 2 (Nrf2) system, which is a master regulator of the anti-oxygen stress response. The pharmacological activation of the Keap1/Nrf2 system is expected to enhance astrocytic protective mechanisms against ROSs, leading to a novel therapeutic strategy for the treatment of Parkinson’s disease.

**Figure 4 ijms-22-06568-f004:**
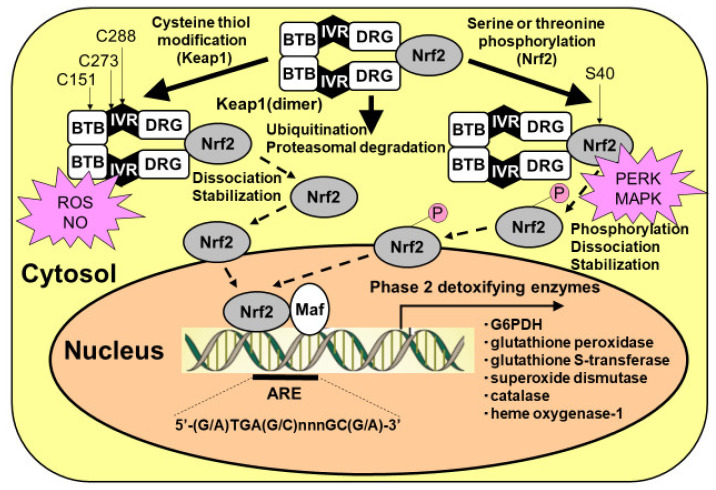
Activation of Kelch-like enoyl-CoA hydratase-associated protein 1 (Keap1)/nuclear factor erythroid 2 p45 subunit-related factor 2 (Nrf2) system in astrocytes. Nrf2 can be activated by at least two mechanisms: (1) the stabilization of Nrf2 via Keap1 cysteine thiol modification, and (2) the phosphorylation of serine residues of Nrf2 by upstream kinases, such as RNA (PKR)-like ER kinase (PERK) and mitogen-activated protein kinase (MAPK). Keap1 presents two characteristic domains, the bric-a-brac, tramtrack, and broad complex (BTB) domain, and the double glycine repeat (DGR) domain. Keap1 bridges the Cullin-3-based E3 ligase and Nrf2 using its BTB and the central intervening region (IVR) to bind Cullin-3 and its DGR to bind Nrf2. The BTB domain participates in Keap1 dimerization. Under basal conditions, Nrf2 is continuously degraded by the ubiquitin/proteasomal system. At least two different mechanisms that facilitate the dissociation of Nrf2 from Keap1, leading to Nrf2 translocation from the cytosol to the nucleus, are known. One involves the modification of thiol residues in the Keap1 protein by reactive oxygen species (ROSs) and nitric oxide (NO). Another mechanism for the dissociation of the Keap1/Nrf2 complex is the phosphorylation of serine residues of Nrf2. Nrf2 then translocates to the nucleus and must heterodimerize with members of the Maf proto-oncogene family to bind to regulatory elements in the DNA and increase antioxidant response element (ARE)-driven transcription.

**Figure 5 ijms-22-06568-f005:**
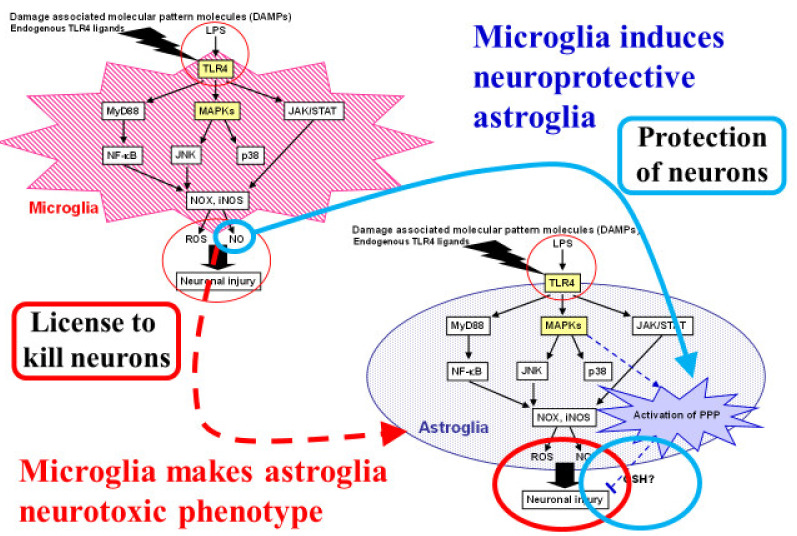
Dual roles of microglia that induce both proinflammatory-neurotoxic and anti-inflammatory-neuroprotective astrocytes. Astrocytes are known to play neurotoxic roles in ischemic stroke as well as neurodegenerative diseases. Microglia induce neurotoxic astrocytes through proinflammatory responses induced by toll-like receptor (TLR) 4 activation. Reactive oxygen species (ROSs) and nitric oxide (NO) also act as neurotoxic molecules that can cause neuronal injury. We focused on the neuroprotective roles of astrocytes through their high glycolysis activity and the pentose-phosphate pathway (PPP) against oxidative stress. Damage-associated molecular patterns (DAMPs) and lipopolysaccharide (LPS), a natural TLR4 ligand, induce ROS and NO production in microglia. We found that NO released from microglia activated astroglial PPP flux through the Kelch-like enoyl-CoA hydratase-associated protein 1 (Keap1)/nuclear factor erythroid 2 p45 subunit-related factor 2 (Nrf2) system. Namely, the NO-induced nitrosylation of Keap1 residues released Nrf2 from Keap1, allowing Nrf2 to act as a transcription factor in an in vitro model of cultured rodent microglia and astroglia.

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
