# Peer review of "Neuroprotective Function of High Glycolytic Activity in Astrocytes: Common Roles in Stroke and Neurodegenerative Diseases"

_ijms, 2021, doi:10.3390/ijms22126568_

Round 1

Reviewer 1 Report

This a BORDERLINE TO REJECTION.

Very interesting field. However, this review is not the first one regarding glycolysis, PPP and astrocytes. So my concern is what's new? What is the originality? The review needs a major revision to be eventually acceptable for publication. I recommend to open the review to an overview integrating Glycolysis, Mitochondria, PPP, Peroxisome, Lipid metabolism, Acetyl-CoA.  The three last are missing and therefore should improve the review. It needs a substantial restructuring. Focusing only on PPP is not enough.

The author has to remove all "unpublished data" or "data not shown". If one unpublished data stay in this review, i will formely reject the manuscript. Only published data have to cited and considered. For any unpublished data; the author has first to officially publish the data in an article.

Finally, when including microglia, the story falls short. Only ROS and NO effects from microglia? What about cytokine effects? 

Reviewer 2 Report

Dear Editor,

The manuscript by Takahashi reviews discussed the roles of astrocytic glycolysis in neuroprotection during stroke and neurodegenerative diseases.

The review is comprehensive, informative and up-to-date (in most parts). Authors were successful in providing some well compiled opinions and summaries. The mechanistic figures and the addition of some suggestions for future directions in the conclusion section can be a good starting point for future studies and will be of interest for IJMS readers and beyond.

However, there is a number of major and minor points that would need to be addressed in order to improve the quality of this paper before it can be accepted for publication.

General:

- This review overlooked some essential and up-to-date work regarding the pathophysiology of stroke and neurodegenerative diseases and recent advances in target validation and future therapies. I have made some suggestions below but authors are encouraged to consider citing updated references throughout the review, whenever possible.

Major:

- Authors need to mention the work by Qian et al. Nature 2020 where they have beautifully shown that the conversion of midbrain astrocytes to dopaminergic neurons, which provide axons to reconstruct the nigrostriatal circuit. Reference:

https://pubmed.ncbi.nlm.nih.gov/32581380/

- The author does not reference a key study from 2020, demonstrating that the development of edema following injury-induced hypoxia is aquaporin-4 dependent. This breakthrough study showed that essential role of targeting astrocytes in CNS injures. Reference to ne included: https://www.cell.com/cell/fulltext/S0092-8674(20)30330-5.

This role has been recently been confirmed by the work of Sylvain et al BBA 2021 which has demonstrated that targeting AQP4 effectively reduces cerebral edema during the early acute phase in in ischemic stroke. They have also shown a link to brain energy metabolism as indicated by the increase of glycogen levels. Reference to be included:

https://pubmed.ncbi.nlm.nih.gov/33561476/

Related to the role of astrocytes through their AQP4. Hypothermia (and hence reduced metabolic activity) has been shown to mediate its effect through affecting the astrocytes. Reference:

https://pubmed.ncbi.nlm.nih.gov/28925524/

https://pubmed.ncbi.nlm.nih.gov/29311824/

https://pubmed.ncbi.nlm.nih.gov/22353781/

- Author needs to briefly discuss future directions following towards the end of their discussion and conclusion. This could include, but not limit to, the use of humanized self-organized models, organoids, 3D cultures and human microvessel-on-a-chip platforms especially those which are amenable for advanced imaging such as TEM and expansion microscopy since they enable real-time monitoring of metabolic activity. This is quite important since the expression of key genes and proteins in astrocytes has proven to be varied between 2D and 3D cultures. References to be included:

https://pubmed.ncbi.nlm.nih.gov/30165870/

https://pubmed.ncbi.nlm.nih.gov/33117784/

https://pubmed.ncbi.nlm.nih.gov/31889243/

-End of discussion and towards the conclusion: Stroke and neurodegenerative diseases are yet incurable diseases. Author needs to point out to the recent advances in applying the use of high-throughput screening and computer-aided drug design as have been nicely reviewed by Aldewachi et al 2021 and Salman et al 2021 as they can provide a novel insight that can support target validation in future studies. References to be included:

https://pubmed.ncbi.nlm.nih.gov/33672148/

https://pubmed.ncbi.nlm.nih.gov/33925236/

Best.

Round 2

Reviewer 1 Report

I do accept the corrections and the manuscript.

Reviewer 2 Report

Dear Editor,

The author has successfully addressed the majority of my comments and concerns in order to improve the quality of the manuscript.

I believe that the corrections, additional sections and updated references, have contributed to enhancing the clarity of the manuscript, which I can now endorse for publication.

All the best!